

# From lowlands to highlands: how elevation and habitat complexity drive anuran multidimensional diversity?

Iuri Ribeiro Dias[1,*], Kássio de Castro Araújo[2,*], Jorge Mario Herrera-Lopera[3], Caio Vinícius de Mira-Mendes[4], Tadeu Teixeira Medeiros[1], Marcos Ferreira Vila Nova[1], Marcelo Felgueiras Napoli[5] and Mirco Solé[1,3,6]

[1] Programa de Pós-Graduação em Zoologia, Departamento de Ciências Biológicas, Universidade Estadual de Santa Cruz, Ilhéus, Bahia, Brazil
[2] Grupo de Pesquisa em Biodiversidade e Biotecnologia do Centro-Norte Piauiense—BIOTECPI, Instituto Federal de Educação, Ciência e Tecnologia do Piauí, Pedro II, Piauí, Brazil
[3] Programa de Pós-Graduação em Ecologia e Conservação da Biodiversidade, Departamento de Ciências Biológicas, Universidade Estadual de Santa Cruz, Ilhéus, Bahia, Brazil
[4] Departamento de Biologia, Universidade Estadual do Maranhão, São Luís, Maranhão, Brazil
[5] Instituto de Biologia, Universidade Federal da Bahia, Salvador, Bahia, Brazil
[6] Museum Koenig Bonn (ZFMK), Leibniz Institute for the Analysis of Biodiversity Change, Bonn, North Rhine-Westphalia, Germany
* These authors contributed equally to this work.

Corresponding author
Iuri Ribeiro Dias,
iurirdias@hotmail.com

## ABSTRACT

**Background:** Montane environments in Neotropical regions are known for their rich diversity of amphibians, but the ecological drivers behind this diversity along altitudinal gradients remain poorly understood. We investigated the effects of the altitudinal range and local environmental variables on the taxonomic, phylogenetic, and functional alpha and beta diversity of anuran assemblages along an altitudinal gradient in the Atlantic Forest of northeastern Brazil.

**Methods:** We characterized the richness, abundance, taxonomic, functional, and phylogenetic diversity of anurans in 24 transects within the interior of the forest along an altitudinal range of 200–950 m in the Private Reserve of Natural Heritage (RPPN) Serra Bonita, southern Bahia state, northeastern Brazil. For each transect, we measured the following environmental variables: altitude, leaf litter depth and cover, canopy opening, number of tank-bromeliads, number of trees, and mean air temperature.

**Results:** We found 36 anuran species distributed in 10 families. Altitudinal strata plays an important role in explaining anuran abundance, with direct-developing frogs being the most abundant species. The number of tank-bromeliads was interpreted as having the most substantial support to explain the anuran abundance, lineage richness and functional diversity, whereas leaf litter depth influenced the dominant lineages. Additionally, altitude significantly influenced taxonomic and phylogenetic dissimilarity. Lastly, we found an inverse pattern of altitudinal Rapoport's rule, in which species with optimal altitudes in the highlands exhibit a lower range-size distribution, likely due to habitat specialization or micro-endemism at higher altitudes.

**Conclusion:** Altitude significantly influenced the abundance, taxonomic composition, and phylogenetic diversity of anuran communities, with higher

elevations supporting a greater number of individuals and distinct evolutionary lineages. In contrast, functional diversity did not vary with altitude, suggesting functional redundancy, where different species perform similar ecological roles, thereby maintaining community resilience. Local factors, such as the number of tank-bromeliads and leaf litter depth, were also key variables shaping community structure. Given the high species turnover and the presence of unique evolutionary lineages, especially in the highlands, conservation efforts should prioritize the protection of the entire montane habitat to sustain the ecological and evolutionary processes that support this exceptional biodiversity. Understanding how species are distributed and identifying the most important filters of anuran diversity along altitudinal gradients in the Atlantic Forest is essential for developing management plans and conservation actions in this threatened region that harbors one of the world's most remarkable assemblages of anurans.

## INTRODUCTION

The distribution of organisms along geographic gradients has aroused interest among biologists since the middle of the 19th century (*Darwin, 1839*; *Von Humboldt, 1849*; *Wallace, 1878*). Environmental conditions along altitudinal gradients might affect the local biota, resulting in fauna and flora zonation (*Ricklefs, 1993*). Hence, how mountains' environmental changes influence species richness, abundance, and composition is a reason for debates and studies in different parts of the world to the present day (*e.g.*, *Rahbek et al., 2019*; *Villacampa et al., 2019*; *Carvalho-Rocha, Peres & Neckel-Oliveira, 2021*; *Liu et al., 2022*).

Historically, altitudinal gradients were supposed to reflect latitudinal patterns as suggested by several authors (*MacArthur, 1972*; *Begon, Harper & Townsend, 1990*; *Stevens, 1992*). Two main patterns of species richness have been widely documented across taxa: (1) a monotonic decrease in species richness with increasing altitude (*Terborgh, 1977*; *Hunter & Yonzon, 1993*), and (2) a unimodal pattern, where species richness peaks at intermediate elevations (*Rahbek, 1995*; *McCain, 2009*; *Grytnes & McCain, 2013*). The latter is often explained by the Mid-Domain effect, which predicts higher species overlap in the middle of a bounded domain (*Colwell & Hurtt, 1994*; *Colwell & Lees, 2000*). However, the drivers of these patterns are complex and multifaceted, involving interactions between climatic, spatial, evolutionary, and biotic factors (*McCain & Grytnes, 2010*). Additionally, there is variation in these patterns across different taxonomic groups. For instance, plants and non-flying small mammals frequently show mid-elevation peaks, while reptiles often exhibit decreasing trends in species richness with increasing altitude (*Grytnes & McCain, 2013*). In contrast, bats display both patterns (monotonic decline and mid-elevation peaks) in similar proportions across studies (*Grytnes & McCain, 2013*).

Because amphibians generally have a complex life cycle, with an aquatic larval stage followed by metamorphosis into an arboreal, semi-aquatic, or terrestrial adult, cutaneous respiration, and inhabiting different microhabitats, they are considered one of the most sensitive groups to environmental changes among vertebrates (*Duellman & Trueb, 1994*; *Wells, 2007*). Thus, they are interesting models for understanding how environmental variables influence the structure and distribution of the community. However, as in most other groups, these patterns are contentious in mountain frogs. For instance, some studies found richness peaks at intermediate altitudinal bands (*Hu et al., 2011*; *Carvalho-Rocha, Peres & Neckel-Oliveira, 2021*), others a monotonic decrease of species richness with increasing altitude (*Khatiwada et al., 2019*; *Siqueira et al., 2021*; *Siqueira, Vrcibradic & Rocha, 2025*), and the absence of a relationship between richness and altitude (*Goyannes-Araújo et al., 2015*; *Araújo et al., 2025*).

Regarding species distribution patterns, one of the most notorious hypotheses to explain the influence of the latitudinal gradient on their distribution is Rapoport's Rule (*Stevens, 1989*). It assumes species from higher latitudes occur in wider latitudinal ranges than species from lower latitudes (*Rapoport, 1975*; *Stevens, 1989*). This hypothesis was also extended (*Stevens, 1992*) and tested for altitudinal gradients for different taxonomic groups (*e.g.*, *Almeida-Neto et al., 2006*; *Kim et al., 2019*; *Araújo et al., 2025*; *Kohlmann, Arriaga-Jiménez & Portela Salomão, 2021*). In amphibians, this assumption also remains an unsolved issue, with different patterns observed regarding their distribution in altitudinal gradients (*e.g.*, *Goyannes-Araújo et al., 2015*; *Khatiwada et al., 2019*; *Chettri & Acharya, 2020*; *Araújo et al., 2025*; *Siqueira et al., 2021*).

Although studies dealing with anurans from mountains in northeastern Brazil have been increasing recently (*e.g.*, *Xavier & Napoli, 2011*; *Roberto et al., 2017*; *Rojas-Padilla et al., 2020*; *Araújo et al., 2025*; *Bastos & Ramos, 2022*), the main drivers of species richness and distribution in most of these altitudinal gradients are still unknown. Among them, the Serra Bonita RPPN (Private Natural Heritage Reserve) complex, in Bahia state, is one of the amphibians' hotspots in the Atlantic Forest (*Dias et al., 2014*), but little is known about the influence of the altitudinal and environmental conditions influencing anuran communities, including their species composition, functional roles, and phylogenetic lineages. Understanding these patterns may provide valuable insights for conservation planning (*Pimm & Brown, 2004*).

Here, we investigate the multidimensional diversity and distribution patterns of anurans along an altitudinal gradient in the Serra Bonita RPPN complex, Bahia state, northeastern Brazil. First, we tested how altitudinal range and local environmental variables influence anuran abundance as well as their taxonomic, functional, and phylogenetic diversity and composition. We expect amphibian diversity to decrease with increasing altitude (*Lomolino, 2001*; *Siqueira & Rocha, 2013*) and hypothesize that anuran abundance, multidimensional diversity, and composition will be related, as frogs are among the most sensitive vertebrates to local environmental dynamics (*Hopkins, 2007*). Finally, we tested whether species distribution follows the predictions of Rapoport's rule for altitudinal ranges, expecting that species with optimal altitudes at high elevations would have a wider

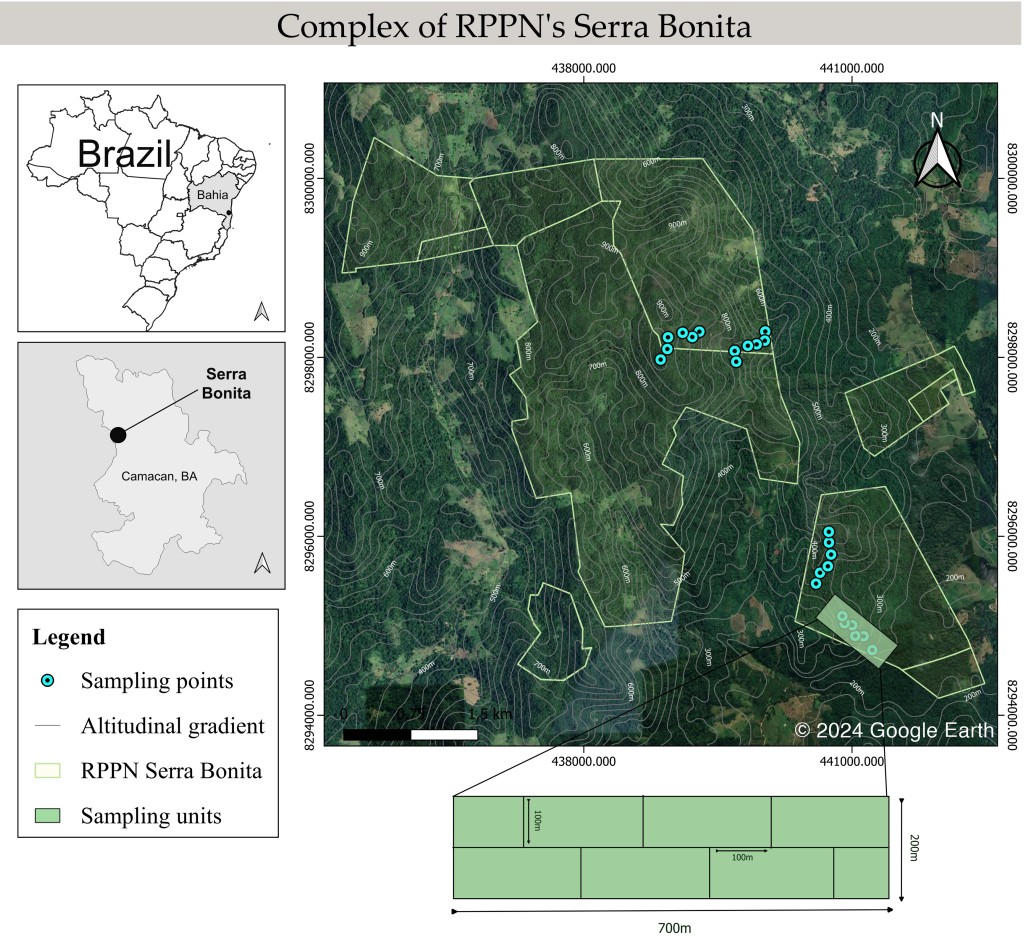

**Figure 1 Schematic map of the Serra Bonita RPPN complex, showing the transects sampled along the altitudinal gradient.** Map built using the Google Satellite tool from QGis 3.34.6 (©2024 Google Earth).

distribution along the gradient due to their greater adaptation to extreme conditions in the highlands (*Stevens, 1992*).

# MATERIALS AND METHODS

## Study area

Anuran sampling was conducted in the Serra Bonita RPPN complex in Camacan and Pau Brasil municipalities, Bahia state, northeastern Brazil (−15.3836 S, −39.5502 W). It is a montane complex covering a total area of 7,500 hectares in the Atlantic Forest with an altitudinal gradient ranging from 200 to 950 m (Fig. 1). The vegetation is composed of a mosaic with different succession stages of secondary forests interspersed with cocoa crops and pastures (see *Dias et al., 2014* for a detailed description of the study area).

## Sampling methods

We conducted monthly field trips over six consecutive days from December 2009 to November 2010. Four altitudinal bands were defined: 200–300 (low), 400–500 (mid),

600–700 (mid-high), and 800–900 (high) m, and six 100 m long linear transects in the forest interior were marked in each band, giving a total of 24 transect sampling locations. A 700 m long main track was marked out within each altitudinal band, and 100 m transects were placed perpendicular to this track (Fig. 1). The installation site of the first transect was determined by a draw within 100 m from the start of the main trail. In addition, we randomly determined the side of the main trail (right or left) where each transect would be installed. Then, the other transects were installed systematically 100 m away from each other and on the opposite side of the previous transect. We did not set up transects near the forest edge or in water bodies to focus on species in the forest interior, with a minimum distance to the edge of 300 m. Sampling was carried out by active visual and acoustic search (*Heyer et al., 1994*; *Rödel & Ernst, 2004*) conducted by two researchers for 40 min in each transect during the night. All transects were inspected once every sampling month.

For each transect, we measured the following environmental variables: altitude, leaf litter depth (LLD) and cover (LLC), canopy opening (OCA), number of tank-bromeliads (BRO), including both epiphytic and terrestrial bromeliads, number of trees (NTR), and air temperature (T). On each transect, five points were marked for collecting the variables (LLC, LLD, and OCA) at 10, 30, 50, 70, and 90 m from the beginning of the transect. At each of these points, a 1 × 1 m plot was established on each side of the trail and assigned values from 0 to 4 (0 = 0–20%; 1 = 20–40%; 2 = 40–60%; 3 = 60–80%; and 4 = 80–100%) to represent the percentage of leaf litter cover. In the center of these plots, we measured the leaf litter depth. A hemispherical photo of the canopy was taken at 1.20 m above ground level at each of the five points marked on the transects to assess the canopy opening. These photos were later analyzed with the Gap Light Analyzer 2.0 program. We counted all bromeliads (up to 5 m in height) and trees (>5 cm in circumference at breast height) that were within 1.5 m on either side of the transect. The air temperature was measured at the beginning and end of each transect sampling with a digital thermometer with an accuracy of 0.5 °C. The air temperature was the average between the temperatures at the beginning and end of each collection.

This research was approved by the Ethical Committee on Animal Use at the Universidade Estadual de Santa Cruz (CEUA-UESC 006/09). The specimens were collected under authorization (ICMBio #13708) granted by Instituto Chico Mendes de Conservação da Biodiversidade (ICMBio/SISBIO) from the Ministério do Meio Ambiente (MMA) of Brazil. Voucher specimens were deposited in the herpetological collection of the Museu de Zoologia da Universidade Estadual de Santa Cruz (MZUESC), Ilhéus, Bahia state, Brazil, and are listed in Appendix I of *Dias et al. (2014)*. Anuran nomenclature and distribution follow *Frost (2024)*, whereas the species conservation status is according to *IUCN (2024)*.

## Phylogeny and functional traits

The phylogeny of the species in this study was constructed from the phylogeny for amphibians available in Time Tree 5 (*Kumar et al., 2022*). Species not found within the base phylogeny were assigned as polytomies using the V.PhyloMaker package in R (*Jin & Qian, 2019*; *R Core Team, 2024*).

We considered the following functional traits for each species: body size (snout-vent length, in mm), habitat preference (forest, open area, or both), calling site (forest floor, lentic waters, lotic waters, shrubs, bromeliads, canopy), reproductive mode (following *Nunes-de-Almeida, Haddad & Toledo, 2021*), habit (arboreal, cryptozoic, phytotelmata, semi-arboreal, and terrestrial) and activity period (diurnal, nocturnal) (Appendix). Data were compiled following *Haddad et al. (2013)* and further complemented with our own in-field observations. These traits were chosen to represent different aspects of the species' interaction with its environment (*González et al., 2016*).

## Data analyses

The species abundance in each transect was determined as the total abundance of species collected for each transect during the sampling duration (cumulative abundance). We assessed the efficiency of our sampling by estimating sampling coverage (SC) (*Chao et al., 2014*). Since comparing biodiversity requires similar sampling coverage across all assemblages (transects), we calculated SC along transects and compared their 95% confidence intervals (*Cumming, Fidler & Vaux, 2007*; *Cultid-Medina & Escobar, 2019*). After confirming that SC values overlapped across all transects (0.86–0.97), we proceeded with the diversity comparisons using observed values (*Chao et al., 2014*).

We estimated the taxonomic, functional, and phylogenetic diversity of anuran species using the Hill number framework (*Chao et al., 2021*). This approach partitions each dimension of diversity into three measures: $^0D$, which represents species, lineage or functional group richness; $^1D$, which corresponds to the effective number of abundant species, lineages or functional groups and reflects overall diversity; and $^2D$, which captures the effective number of highly abundant or dominant species, lineages or functional groups. Since a higher number of dominant species indicates greater evenness in abundance distribution, $^2D$ may also be interpreted as a measure of species, lineages or functional groups evenness (*Hill, 1973*; *Jost, 2006*). These estimations were carried out using the iNEXT.3D package of R (*Chao, 2024*; *R Core Team, 2024*).

We used abundance and multidimensional diversity data to fit generalized linear models (GLMs), with Altitude, LLD, LLC, OCA, BRO, NTR, and T as explanatory variables. For discrete variables (*i.e.*, abundance and species richness—$^0TD$), we fitted Poisson and negative binomial models, while for continuous variables (*i.e.*, all other response variables), we fitted Gaussian and Gamma models with a logarithmic link function (*Buckley, 2015*). In all cases, we first fitted a global model including all variables, ensuring that residuals were not overdispersed, exhibited no heteroscedasticity patterns or independence violations, and followed a uniform distribution when simulated (*Hartig, 2016*). We then removed variables with a variance inflation factor (VIF) exceeding 10 (*Zuur et al., 2009*). If the confidence intervals of the VIF values for two or more high VIF variables overlapped, we tested alternative global models by removing one variable at a time and compared them using the corrected Akaike information criterion for small samples (AICc), retaining the model with the lowest AICc value. From the global model, we identified the top models—those combinations of explanatory variables that provided the best trade-off between model fit and complexity—using the *dredge* function from the

MuMin package (*Bartoń, 2010*). The top models were defined as those with a ΔAICc < 2. When multiple models fell within this threshold, we selected the one with the fewest parameters as a pragmatic choice among equally supported alternatives (*Zuur et al., 2009*; *Burnham & Anderson, 2010*; *Richards, 2015*). If the null model was within this subset, we favored it, as it represents the simplest possible explanation consistent with the data—namely, randomness (*Richards, 2015*). Model residuals were assessed using the performance package in R (*Lüdecke et al., 2019*; *R Core Team, 2024*). For all cases in which the selected model was not the null model, we assessed the presence of spatial correlation in the residuals using Moran's test (*Negrete-Yankelevich & Fox, 2015*).

To assess differences in species composition, functional groups, and lineages across transects, we followed the approach proposed by *Cardoso et al. (2014a)*. This method extends the Jaccard index (*Carvalho et al., 2013*) to account for phylogenetic and functional traits, yielding three specific versions: taxonomic beta (Tβ), equal to Jaccard index, phylogenetic beta (Pβ) and functional beta (Fβ) (*Cardoso et al., 2014b*). To assess how environmental variables influence dissimilarity between transects, we fitted generalized dissimilarity models (GDMs, *Ferrier et al., 2007*), incorporating the same explanatory variables as in the GLMs, along with transect geographic distance (in meters). The importance of each variable in the GDMs was evaluated by sequentially removing them and refitting the model. Model validity was assessed based on its *p*-value, the percentage of deviance explained, and the explanatory power in cross-validation tests (1,000 iterations) (*Mokany et al., 2022*). Dissimilarity analyses were performed using the BAT package, while GDMs were fitted with the gdm package, all within the R environment (*Cardoso et al., 2014a*; *Fitzpatrick et al., 2015*; *R Core Team, 2024*). The R code used to perform the alpha and beta multidimensional diversity analyses, as well as the datasets for this work, is available in the Supplemental Files.

To understand the anuran species distribution patterns in the studied mountain, we measured the maximum and minimum altitude where each anuran species was recorded to estimate their range-size distribution (the highest altitude minus the lowest altitude where each species was recorded). We give a range of 100 m to species recorded at a single sampling point (*Kim et al., 2019*). The optimal altitude of each species might be understood as the local where it has a maximum abundance (*Whittaker, 1967*); thus, we used two methods to calculate it: the average of the altitudinal range of each species (see *Stevens, 1992*) and the "Specimen method" (to consult *Almeida-Neto et al., 2006* for more details about formulas and methods). Then, we first assessed the normality of the data distribution using the Shapiro-Wilk test and evaluated the homoscedasticity with the Fligner-Killeen test. As the residuals of our data did not meet the assumptions of normality and homoscedasticity required for simple linear regression, we opted for the non-parametric Kernel regression test (*Nadaraya, 1964*; *Watson, 1964*) to investigate if the anuran distribution along the mountain studied follows Rapoport's rule using the three methods cited above. These analyses were performed using the R packages mgcv (*Wood, 2023*) and vegan (*Oksanen et al., 2016*).

## RESULTS

We registered 1949 individuals belonging to 36 anuran species (Table 1) nested in the following ten families (number of species in parentheses): Brachycephalidae (3), Bufonidae (2), Craugastoridae (1), Eleutherodactylidae (1), Hemiphractidae (1), Hylidae (21), Leptodactylidae (1), Microhylidae (1), Odontophrynidae (1), and Strabomantidae (4). Of these species, about 80% are restricted to the Atlantic Forest, and *Brachycephalus pulex* is listed as endangered (EN), while *Bokermannohyla lucianae* is considered vulnerable (VU) to extinction (*IUCN, 2024*). The dominant species were *Pristimantis vinhai* ($n = 781$), *Haddadus binotatus* ($n = 197$) and *Pristimantis* sp. 1 ($n = 195$), all of which are direct-developing species. In contrast, the other seven species (*Aplastodiscus ibirapitanga*, *Boana pombali*, *Chiasmocleis crucis*, *Dendropsophus novaisi*, *Ischnocnema verrucosa*, and *Physalaemus erikae*) had only one individual recorded each. Transect sampling coverage ranged from 0.86 to 0.97, indicating a relatively high level of sampling completeness (Table 2).

Detailed data on abundance and taxonomic, functional, and phylogenetic diversity values for each transect are presented in Table 2. Transect abundance was significantly and positively associated with altitude ($p$-value = 0.003, Fig. 2A) and the number of tank bromeliads (BRO, $p$-value = 0.009, Fig. 2B). In contrast, for taxonomic diversity ($^q$TD), the null model was selected in all cases ($\Delta$AICc < 2, Figs. 2C–2E). Although alternative models included explanatory variables with significant associations, they were not clearly better supported than the null model, suggesting weak evidence for the effect of those variables. For lineage richness ($^0$PD), we found a significant positive relationship with the number of tank bromeliads ($p$-value = 0.007, Fig. 2F). For lineage diversity ($^1$PD), the null model was the selected model (Fig. 2G). Finally, the number of dominant lineages exhibited a significant positive association with leaf litter depth (LLD, $p$-value: 0.036, Fig. 2H). For functional diversity ($^q$FD), communities with greater richness ($^0$FD), diversity ($^1$FD) and greater number of dominant ($^2$FD) functional groups, were related to a greater number of tank bromeliads ($p$-value < 0.05 for all cases, Figs. 2I–2K). Table 3 presents the metrics for all models with a $\Delta$AICc < 2, for each response variable. None of the selected alpha diversity models, different of null model, showed evidence of spatial correlation in the residuals (Moran I < 0.3, $p$-value > 0.05 in all cases).

The average species composition dissimilarity between transect pairs was 0.58, while phylogenetic and functional composition dissimilarities were 0.47 and 0.50, respectively. Taxonomic dissimilarity was significantly and positively associated with altitude ($p$-value = 0.003), with altitude contributing almost monotonically to taxonomic differentiation among transects (Fig. 3A). Similarly, phylogenetic dissimilarity was significantly and positively correlated with altitude ($p$-value = 0.006), with this effect being particularly pronounced among transect pairs above 600 m in elevation (Fig. 3B). Additionally, we detected a marginally significant relationship between phylogenetic dissimilarity and geographical distance ($p$-value = 0.092, Fig. 3C). In contrast, functional dissimilarity between transects showed no significant association with any of the

**Table 1 Anuran species found in the Serra Bonita RPPN complex, Bahia state, northeastern Brazil.**

| Taxa | IUCN | Distribution |
|---|---|---|
| BRACHYCEPHALIDAE | | |
| *Brachycephalus pulex* Napoli, Caramaschi, Cruz, and Dias, 2011 | EN | At |
| *Ischnocnema* sp. (gr. *parva*) | NA | ? |
| *Ischnocnema verrucosa* (Reinhardt and Lütken, 1862) | LC | At |
| BUFONIDAE | | |
| *Rhinella crucifer* (Wied-Neuwied, 1821) | LC | At |
| *Rhinella hoogmoedi* Caramaschi and Pombal, 2006 | LC | At |
| CRAUGASTORIDAE | | |
| *Haddadus binotatus* (Spix, 1824) | LC | At |
| ELEUTHERODACTYLIDAE | | |
| *Adelophryne* sp. | NA | ? |
| HEMIPHRACTIDAE | | |
| *Gastrotheca pulchra* Caramaschi and Rodrigues, 2007 | LC | At |
| HYLIDAE | | |
| *Aplastodiscus ibirapitanga* (Cruz, Pimenta, and Silvano, 2003) | LC | At |
| *Aplastodiscus weygoldti* (Cruz and Peixoto, 1987) | LC | At |
| *Boana crepitans* (Wied-Neuwied, 1824) | LC | W |
| *Boana faber* (Wied-Neuwied, 1821) | LC | At |
| *Boana pombali* (Caramaschi, Pimenta, and Feio, 2004) | LC | At |
| *Bokermannohyla circumdata* (Cope, 1871) | LC | At |
| *Bokermannohyla lucianae* (Napoli and Pimenta, 2003) | VU | At |
| *Dendropsophus anceps* (Lutz, 1929) | LC | At |
| *Dendropsophus novaisi* (Bokermann, 1968) | LC | At, Ce |
| *Ololygon strigilata* (Spix, 1824) | LC | At |
| *Phasmahyla spectabilis* Cruz, Feio, and Nascimento, 2008 | LC | At |
| *Phyllodytes maculosus* Cruz, Feio, and Cardoso, 2007 | LC | At |
| *Phyllodytes melanomystax* Caramaschi, Silva, and Britto-Pereira, 1992 | LC | At |
| *Phyllodytes wuchereri* (Peters, 1873) | LC | At |
| *Phyllodytes* sp. | NA | ? |
| *Phyllodytes magnus* Dias et al. 2020 | LC | At |
| *Phyllodytes megatympanum* Marciano, Lantyer-Silva, and Solé, 2017 | LC | At |
| *Phyllomedusa burmeisteri* Boulenger, 1882 | LC | At |
| *Scinax eurydice* (Bokermann, 1968) | LC | At |
| *Trachycephalus mesophaeus* (Hensel, 1867) | LC | At |
| *Trachycephalus nigromaculatus* Tschudi, 1838 | LC | At, Ce |
| LEPTODACTYLIDAE | | |
| *Physalaemus erikae* Cruz and Pimenta, 2004 | LC | At |
| MICROHYLIDAE | | |
| *Chiasmocleis crucis* Caramaschi and Pimenta, 2003 | LC | At |
| ODONTOPHRYNIDAE | | |
| *Proceratophrys schirchi* (Miranda-Ribeiro, 1937) | LC | At |

(*Continued*)
| Table 1 (continued) | | |
| --- | --- | --- |
| **Taxa** | **IUCN** | **Distribution** |
| STRABOMANTIDAE | | |
| *Bahius bilineatus* (Bokermann, 1975) | LC | At |
| *Pristimantis vinhai* (Bokermann, 1975) | LC | At |
| *Pristimantis* sp. 1 | NA | ? |
| *Pristimantis* sp. 2 | NA | ? |

Note:
IUCN conservation status: LC, least concern; EN, endangered; VU, vulnerable; and NA, not applicable; and distribution in Brazilian biomes: At, Atlantic forest; Ce, Cerrado; and W, wide distribution.

**Table 2 Sampling coverage (SC), abundance, and taxonomic ($^qTD$), phylogenetic ($^qPD$), and functional ($^qFD$) diversity metrics of anurans across the transects analyzed in this study.**

| Transect | SC | Altitude | Abundance | $^0TD$ | $^1TD$ | $^2TD$ | $^0PD$ | $^1PD$ | $^2PD$ | $^0FD$ | $^1FD$ | $^2FD$ |
| --- | --- | --- | --- | --- | --- | --- | --- | --- | --- | --- | --- | --- |
| 21 | 0.95 | 238 | 79 | 12 | 7.08 | 5.52 | 3.96 | 2.00 | 1.57 | 5.64 | 2.99 | 2.46 |
| 22 | 0.86 | 282 | 55 | 13 | 5.80 | 3.51 | 4.18 | 1.91 | 1.46 | 5.02 | 2.66 | 1.98 |
| 23 | 0.88 | 293.5 | 42 | 10 | 4.94 | 3.38 | 3.49 | 1.86 | 1.48 | 4.44 | 2.76 | 2.31 |
| 24 | 0.94 | 300 | 47 | 8 | 5.44 | 4.57 | 3.27 | 2.01 | 1.60 | 4.47 | 3.02 | 2.74 |
| 25 | 0.98 | 284 | 79 | 8 | 3.25 | 2.10 | 2.97 | 1.58 | 1.31 | 3.50 | 1.90 | 1.55 |
| 26 | 0.95 | 299 | 64 | 10 | 6.83 | 5.85 | 3.77 | 2.06 | 1.61 | 4.26 | 2.91 | 2.63 |
| 41 | 0.94 | 400 | 48 | 9 | 5.06 | 3.74 | 3.89 | 1.98 | 1.52 | 4.00 | 2.91 | 2.50 |
| 42 | 0.9 | 410 | 38 | 9 | 5.69 | 4.38 | 3.69 | 1.98 | 1.52 | 4.85 | 3.04 | 2.41 |
| 43 | 0.92 | 415 | 59 | 13 | 8.14 | 6.18 | 5.85 | 2.38 | 1.68 | 5.38 | 3.40 | 2.74 |
| 44 | 0.93 | 421.5 | 54 | 11 | 6.54 | 4.81 | 3.85 | 1.98 | 1.56 | 4.41 | 2.79 | 2.37 |
| 45 | 0.9 | 449 | 61 | 13 | 8.13 | 6.32 | 4.37 | 2.12 | 1.60 | 5.04 | 3.34 | 2.83 |
| 46 | 0.94 | 463 | 46 | 9 | 6.21 | 5.16 | 3.20 | 1.99 | 1.58 | 4.23 | 3.04 | 2.64 |
| 61 | 0.99 | 619.5 | 77 | 9 | 5.37 | 3.94 | 3.63 | 1.93 | 1.49 | 4.10 | 2.62 | 2.19 |
| 62 | 0.97 | 608.5 | 65 | 9 | 3.10 | 1.87 | 3.53 | 1.64 | 1.33 | 4.04 | 2.09 | 1.65 |
| 63 | 0.97 | 684.5 | 77 | 10 | 6.72 | 5.32 | 3.65 | 2.08 | 1.59 | 3.81 | 2.89 | 2.55 |
| 64 | 0.98 | 672 | 91 | 9 | 3.97 | 2.51 | 3.70 | 1.78 | 1.42 | 4.27 | 2.27 | 1.85 |
| 65 | 0.88 | 694 | 42 | 10 | 5.81 | 4.06 | 3.36 | 1.92 | 1.52 | 3.85 | 2.67 | 2.31 |
| 66 | 0.92 | 698 | 51 | 11 | 5.50 | 3.36 | 3.77 | 1.94 | 1.51 | 5.26 | 2.97 | 2.32 |
| 81 | 0.97 | 838 | 98 | 14 | 8.76 | 6.82 | 4.62 | 2.16 | 1.62 | 5.14 | 3.67 | 3.25 |
| 82 | 0.97 | 896 | 143 | 15 | 7.23 | 4.59 | 5.46 | 2.11 | 1.53 | 5.77 | 3.61 | 2.85 |
| 83 | 0.97 | 910 | 139 | 10 | 2.62 | 1.68 | 4.11 | 1.53 | 1.24 | 4.46 | 1.92 | 1.50 |
| 84 | 0.98 | 933.5 | 160 | 12 | 4.44 | 3.24 | 4.70 | 1.75 | 1.38 | 4.53 | 2.78 | 2.36 |
| 85 | 0.97 | 919 | 211 | 15 | 5.82 | 4.12 | 5.20 | 1.92 | 1.49 | 5.36 | 3.09 | 2.71 |
| 86 | 0.98 | 913 | 123 | 13 | 7.75 | 5.90 | 4.95 | 2.23 | 1.57 | 5.01 | 3.74 | 3.37 |

environmental variables analyzed ($p$-value $> 0.05$ for all cases). Table 4 presents the metrics for all fitted GDMs.

Additionally, we found a significant influence of the optimal altitude of each anuran species on their range-size distribution considering both Stevens's midpoint method

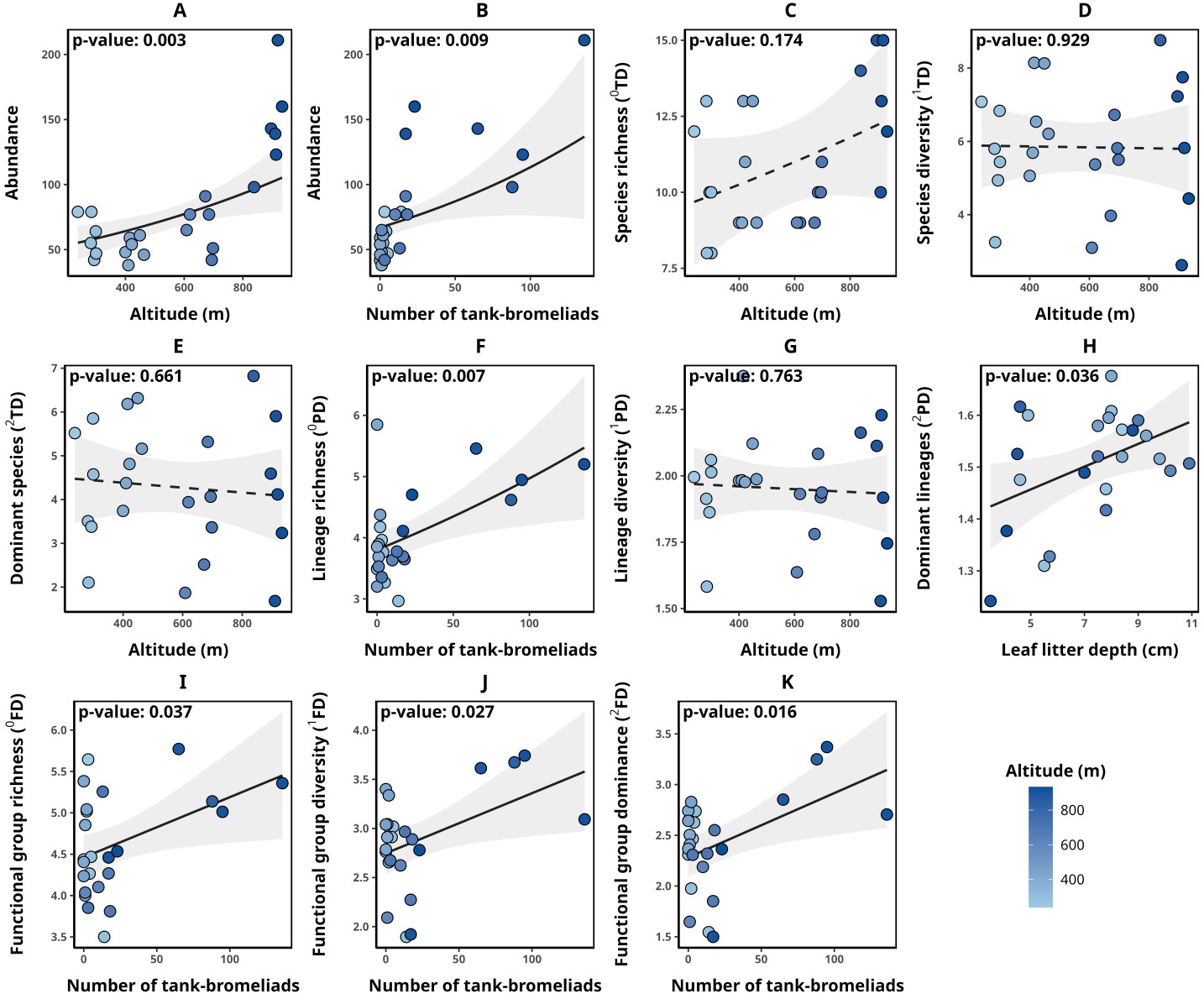

**Figure 2  Relationships between environmental variables and alpha diversity of anurans in Serra Bonita.** Models describing the relationships between environmental variables and (A, B) anuran abundance, (C) species richness, (D) species diversity, (E) number of dominant species, (F) lineage richness, (G) lineage diversity, (H) number of dominant lineages, (I) functional group richness, (J) functional group diversity, and (K) number of dominant functional groups. Solid lines indicate statistically significant relationships ($\alpha < 0.05$), while dashed lines represent non-significant relationships where the null model was the most parsimonious. (A) Illustrates the relationship between altitude and abundance while keeping the number of tank bromeliads constant (21.58 bromeliads); (B) the relationship between the number of tank bromeliads and abundance while maintaining a constant altitude (568.38 m).

($F = 21.84$, $r^2$ (adj.) = 0.788, $p =< 0.0001$) and the Specimen method ($F = 13.62$, $r^2$ (adj.) = 0.726, $p =< 0.0001$). Therefore, anurans with optimal altitudes in the highlands exhibited a lower range-size distribution (Fig. 4).

**Table 3 Top models (ΔAICc < 2) for each alpha diversity response variable.**

**Abundance**

| Model—Distribution family: Negative binomial | k | AICc | ΔAICc | Weight |
|---|---|---|---|---|
| Altitude + Number of tank-bromeliads + Leaf litter depth | 5 | 220.90 | 0.00 | 0.27 |
| Altitude + Number of tank-bromeliads* | 4 | 221.13 | 0.23 | 0.24 |
| Altitude + Number of tank-bromeliads + Leaf litter depth + Number of trees | 6 | 221.46 | 0.56 | 0.20 |
| Altitude + Number of tank-bromeliads + Leaf litter coverage + Leaf litter depth | 6 | 221.64 | 0.73 | 0.19 |
| Altitude + Number of tank-bromeliads + Leaf litter coverage + Leaf litter depth + Number of trees | 7 | 222.82 | 1.92 | 0.10 |

**Species richness ($^0D$)**

| Model—Distribution family: Poisson | k | AICc | ΔAICc | Weight |
|---|---|---|---|---|
| Number of tank-bromeliads | 2 | 111.62 | 0.00 | 0.68 |
| Null model* | 1 | 113.11 | 1.49 | 0.32 |

**Species diversity—effective number of abundant species ($^1D$)**

| Model—Distribution family: Gaussian | k | AICc | ΔAICc | Weight |
|---|---|---|---|---|
| Null model* | 2 | 94.91 | 0.00 | 0.13 |
| Number of tank-bromeliads + Leaf litter depth | 4 | 94.96 | 0.05 | 0.13 |
| Leaf litter coverage | 3 | 95.06 | 0.16 | 0.12 |
| Leaf litter depth | 3 | 95.41 | 0.51 | 0.10 |
| Number of tank-bromeliads + Air temperature | 4 | 95.51 | 0.60 | 0.10 |
| Number of tank-bromeliads | 3 | 95.53 | 0.62 | 0.09 |
| Number of tank-bromeliads + Leaf litter depth + Air temperature | 5 | 95.87 | 0.96 | 0.08 |
| Canopy opening | 3 | 96.06 | 1.16 | 0.07 |
| Number of tank-bromeliads + Leaf litter coverage + Air temperature | 5 | 96.30 | 1.39 | 0.06 |
| Leaf litter coverage + Canopy opening | 4 | 96.39 | 1.48 | 0.06 |
| Number of tank-bromeliads + Leaf litter coverage | 4 | 96.62 | 1.71 | 0.05 |

**Dominant species—effective number of dominant species ($^2D$)**

| Model—Distribution family: Gaussian | k | AICc | ΔAICc | Weight |
|---|---|---|---|---|
| Null model * | 2 | 89.02 | 0.00 | 0.15 |
| Leaf litter coverage | 3 | 89.05 | 0.02 | 0.15 |
| Number of tank-bromeliads + Air temperature | 4 | 89.64 | 0.61 | 0.11 |
| Leaf litter depth | 3 | 89.82 | 0.79 | 0.10 |
| Number of tank-bromeliads + Leaf litter coverage + Air temperature | 5 | 89.93 | 0.90 | 0.10 |
| Number of tank-bromeliads | 3 | 90.53 | 1.50 | 0.07 |
| Canopy opening | 3 | 90.54 | 1.52 | 0.07 |
| Number of tank-bromeliads + Leaf litter depth | 4 | 90.74 | 1.71 | 0.06 |
| Leaf litter coverage + Canopy opening | 4 | 90.76 | 1.74 | 0.06 |
| Number of tank-bromeliads + Leaf litter depth + Air temperature | 5 | 90.78 | 1.76 | 0.06 |
| Leaf litter coverage + Air temperature | 4 | 90.91 | 1.88 | 0.06 |

**Lineage richness ($^0PD$)**

| Model—Distribution family: Gamma (link = "log") | k | AICc | ΔAICc | Weight |
|---|---|---|---|---|
| Number of tank-bromeliads* | 3 | 48.40 | 0.00 | 0.63 |
| Number of tank-bromeliads + Leaf litter coverage | 4 | 49.43 | 1.03 | 0.37 |

**Lineage diversity—effective number of abundant lineages ($^1$PD)**

| Model—Distribution family: Gaussian | k | AICc | ΔAICc | Weight |
| --- | --- | --- | --- | --- |
| Leaf litter depth | 3 | −7.51 | 0.00 | 0.18 |
| Leaf litter depth + Canopy opening | 4 | −7.15 | 0.36 | 0.15 |
| Number of tank-bromeliads + Leaf litter depth | 4 | −6.98 | 0.53 | 0.14 |
| Canopy opening | 3 | −6.46 | 1.06 | 0.11 |
| Null model* | 2 | −6.32 | 1.20 | 0.10 |
| Number of tank-bromeliads + Leaf litter depth + Air temperature | 5 | −5.97 | 1.54 | 0.08 |
| Leaf litter coverage + Canopy opening | 4 | −5.88 | 1.64 | 0.08 |
| Leaf litter coverage | 3 | −5.82 | 1.69 | 0.08 |
| Number of tank-bromeliads + Leaf litter depth + Canopy opening | 5 | −5.76 | 1.76 | 0.08 |

**Dominant lineages—effective number of highly abundant lineages ($^2$PD)**

| Model—Distribution family: Gaussian | k | AICc | ΔAICc | Weight |
| --- | --- | --- | --- | --- |
| Leaf litter depth* | 3 | −38.01 | 0.00 | 0.72 |
| Leaf litter depth + Canopy opening | 4 | −36.15 | 1.86 | 0.28 |

**Functional-group richness ($^0$FD)**

| Model—Distribution family: Gaussian | k | AICc | ΔAICc | Weight |
| --- | --- | --- | --- | --- |
| Number of tank-bromeliads* | 3 | 46.22 | 0 | 1 |

**Functional-group diversity—effective number of abundant functional groups ($^1$FD)**

| Model—Distribution family: Gaussian | k | AICc | ΔAICc | Weight |
| --- | --- | --- | --- | --- |
| Number of tank-bromeliads* | 3 | 34.93 | 0 | 0.4 |
| Number of tank-bromeliads + Leaf litter depth | 4 | 35.68 | 0.74 | 0.27 |
| Number of tank-bromeliads + Air temperature | 4 | 36.46 | 1.53 | 0.18 |
| Number of tank-bromeliads + Number of trees | 4 | 36.93 | 1.99 | 0.15 |

**Dominant functional-groups—effective number of dominant functional groups ($^2$FD)**

| Model—Distribution family: Gaussian | k | AICc | ΔAICc | Weight |
| --- | --- | --- | --- | --- |
| Number of tank-bromeliads* | 3 | 32.03 | 0 | 0.39 |
| Number of tank-bromeliads + Leaf litter depth | 4 | 32.95 | 0.92 | 0.25 |
| Number of tank-bromeliads + Air temperature | 4 | 33.24 | 1.21 | 0.21 |
| Number of tank-bromeliads + Number of trees | 4 | 33.93 | 1.9 | 0.15 |

**Note:**
The model with the fewest parameters was selected (indicated with *). k, Number of parameters; AICc, Akaike information criterion corrected for small sample sizes; ΔAICc, difference in AICc relative to the best-supported model; weight, model weight.

## DISCUSSION

Our results revealed that altitude plays a significant role in structuring anuran communities along the altitudinal gradient in the Atlantic Forest of southern Bahia, affecting abundance, taxonomic composition, and phylogenetic diversity. Frog abundance increased with altitude, but we found no significant relationship between species richness and altitude. This pattern contrasts with previous studies that frequently report a decline in species richness and abundance with increasing altitude in the Atlantic Forest, or a peak in richness at intermediate altitudes (*e.g.*, *Carvalho-Rocha, Peres & Neckel-Oliveira, 2021*;

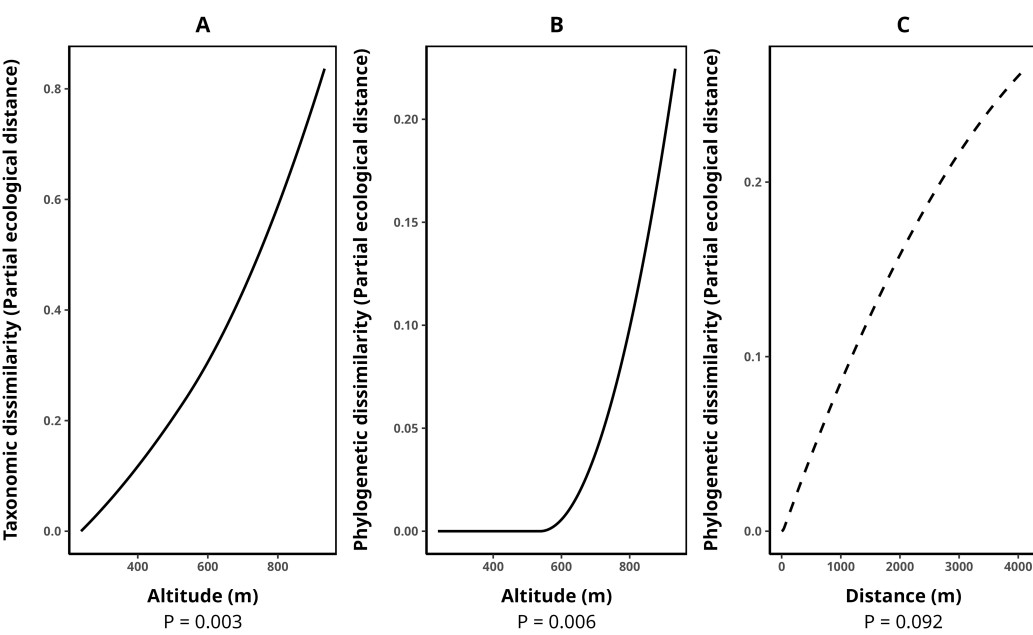

**Figure 3 Splines illustrating the relationships between environmental variables and transect dissimilarity, as predicted by generalized dissimilarity models (GDMs).** The peak values of each spline indicate the total dissimilarity explained by a given variable while holding all others constant. The shape of the spline represents how dissimilarity varies along the environmental gradient. (A) Relationship between altitude and taxonomic dissimilarity, (B) relationship between altitude and phylogenetic dissimilarity, and (C) relationship between geographic distance (m) and phylogenetic dissimilarity. Solid lines denote statistically significant relationships ($\alpha < 0.05$), while dashed lines indicate marginally significant relationships ($p$-value between 0.05 and 0.1). Relationships with functional dissimilarity were not significant and, therefore, were not included in the plot. Dissimilarities were calculated using the taxonomic, phylogenetic, and functional versions of the Jaccard index.

**Table 4 Specification and metrics of the dissimilarity models (GDMs) for taxonomic, phylogenetic, and functional beta diversity (Jaccard-based—$\beta_{cc}$ distances) between transects.**

| Response variable | Model $p$-value | Explained deviance (%) | Explained cross-validation (%) | Predictor | $p$-value |
|---|---|---|---|---|---|
| **Taxonomic beta (T$\beta$)** | 0.00 | 51.24 | 34.67 | **Altitude** | **0.00** |
| **Phylogenetic beta (P$\beta$)** | 0.00 | 46.26 | 36.93 | **Altitude** | **0.00** |
| | | | | *Geographic distance* | *0.09* |
| **Functional beta (F$\beta$)** | 0.00 | 30.24 | 9.44 | Altitude | 0.39 |
| | | | | Geographic distance | 0.19 |
| | | | | Canopy opening | 0.62 |
| | | | | Number of tank-bromeliads | 0.36 |
| | | | | Air temperature | 0.72 |
| | | | | Number of trees | 0.69 |

**Note:**
Variables in bold indicate statistically significant relationships ($p$-value < 0.05), while italicized variables indicate marginally significant relationships (0.05 > $p$-value < 0.1).

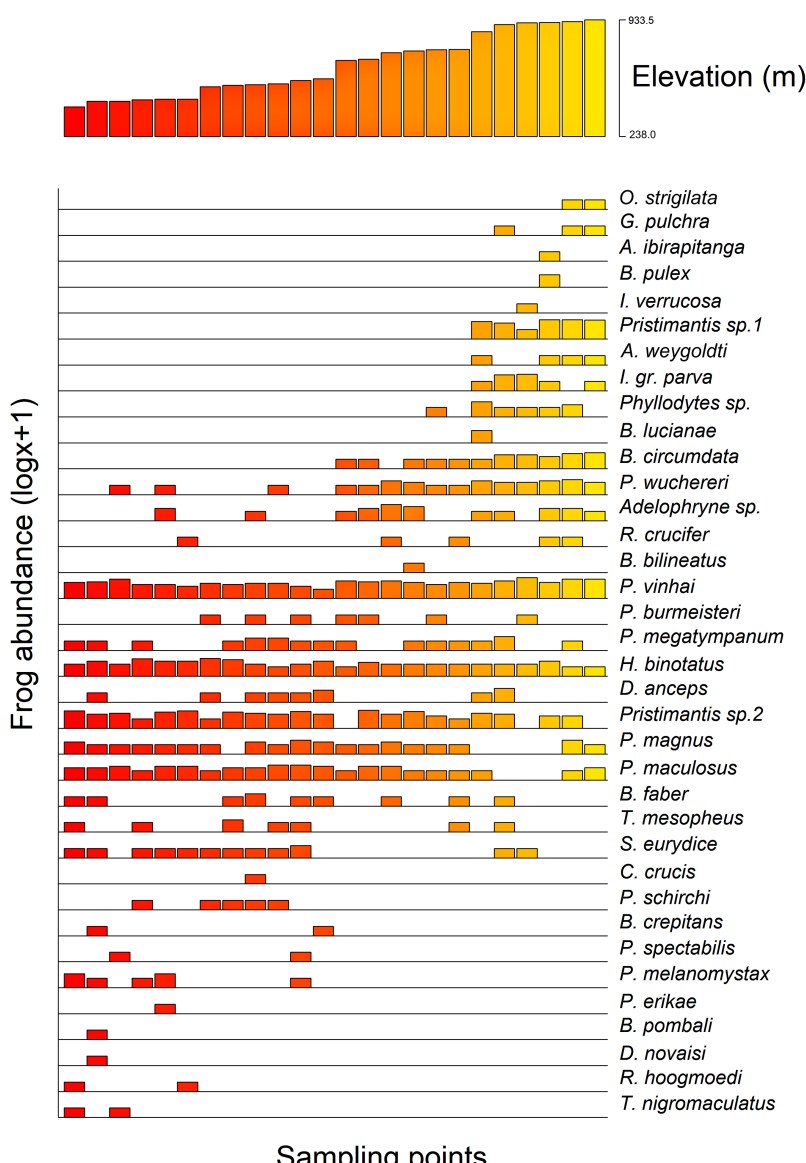

**Figure 4 Anurans' species occurrence and abundance along the altitudinal gradient studied in the Serra Bonita RPPN complex, state of Bahia, northeastern Brazil.** (A) The upper panel shows the altitudinal distribution of each sampled point, and (B) the lower panel shows the abundance of observed species (log x + 1 for better visualization) at each point.

*Matavelli et al., 2022*; *Santos-Pereira & Rocha, 2025*; *Siqueira et al., 2021*; *Siqueira, Vrcibradic & Rocha, 2025*). In addition to altitude, local environmental variables, such as the number of tank bromeliads and leaf litter depth, were key factors in structuring the communities. This finding is consistent with studies highlighting the importance of local factors, such as the availability of specific microhabitats, in maintaining diversity along altitudinal gradients (*e.g.*, *Siqueira et al., 2021*; *Carvalho-Rocha, Peres & Neckel-Oliveira, 2021*; *Wang et al., 2022*; *Zhao et al., 2022*). Furthermore, we observed an inverse pattern to Rapoport's rule, with species at higher altitudes exhibiting narrower range-size
distributions, likely reflecting habitat specialization or microendemism. These results underscore the complexity of diversity patterns along altitudinal gradients and emphasize the need for multidimensional approaches to understand the mechanisms underlying the structuring of these communities.

We recorded 36 anuran species by sampling strictly forested environments, excluding lentic or lotic water bodies. Most herpetofaunistic studies on anurans have focused on areas near ponds, where most species rely on water for reproduction (*Wells, 2007*). However, anurans also depend on forested environments for essential resources such as food and shelter (*Duellman & Trueb, 1994*). By sampling in the forested environments of the Serra Bonita RPPN complex, we recorded almost half of the anuran fauna known for this mountain (*Dias et al., 2014*). In addition, the anurans' species richness found in this study aligns with that reported in other checklists for forested environments of the Atlantic Forest (*Rojas-Padilla et al., 2020*; *Protázio et al., 2021*; *Lima et al., 2021*; *Siqueira et al., 2022*; *Souza-Costa et al., 2024*), highlighting the critical role of natural forests in maintaining anuran diversity.

## Influence of altitude on multidimensional alpha diversity of anurans

Our findings do not support a relationship between species richness and altitude, which contrasts with the commonly observed patterns of either a mid-elevation peak or a monotonic decline in montane amphibian communities (*Carvalho-Rocha, Peres & Neckel-Oliveira, 2021*; *Matavelli et al., 2022*; *Dahl et al., 2024*; *Khatiwada et al., 2019*; *Bassetto et al., 2024*; *Siqueira et al., 2021*; *Siqueira, Vrcibradic & Rocha, 2025*). In tropical mountain ecosystems, species richness generally decreases at higher elevations due to harsh environmental conditions, such as lower temperatures, reduced primary productivity, limited available area, and resource scarcity (*Rahbek, 1995*, *2005*; *McCain, 2005*, *2009*). However, in some low-elevation montane systems (~1,000–1,200 m), species richness may increase with altitude (*Naniwadekar & Vasudevan, 2007*), or remain relatively stable across the gradient (*Goyannes-Araújo et al., 2015*), as observed also in our study. This pattern of high species richness in low-elevation montane systems may be more prevalent in mountains below 1,000 m than previously recognized in the literature. Most studies examining broader altitudinal ranges, which often report a hump-shaped distribution for anurans, describe an initial increase in species richness up to approximately 1,000 m, followed by a decline at higher elevations (*Carvalho-Rocha, Peres & Neckel-Oliveira, 2021*; *Matavelli et al., 2022*; *Dahl et al., 2024*). Thus, in moderately elevated tropical mountains, environmental conditions may remain conducive to supporting amphibian diversity even at upper elevations.

While most studies report a decline in anuran abundance with increasing elevation (*Khatiwada & Haugaasen, 2015*; *Khatiwada et al., 2019*; *Villacampa et al., 2019*; *Carvalho-Rocha, Peres & Neckel-Oliveira, 2021*), our findings reveal a contrasting pattern: a significant positive relationship between altitude and anuran abundance, with the highlands harboring the highest number of individuals. This result may be linked to local factors in our study area, particularly the increased availability of tank bromeliads with altitude, which emerged as a key predictor of anuran abundance (see discussion below).

Additionally, since our study area does not exceed 1,000 m in elevation, the uppermost sites still provide favorable conditions for the persistence and proliferation of species, particularly those with direct development, mitigating the restrictive effects of extreme cold observed at higher altitudes. Up to this altitudinal range, we hypothesize that anuran communities still benefit from milder temperatures and high humidity, reducing physiological stress and promoting greater frog abundance. However, a comprehensive meta-analysis and future studies involving mountainous areas with a higher altitudinal range may shed more light on this issue in the future.

The highest abundance of Terrarana frogs, such as brachycephalids, craugastorids, and strabomantids, might be associated with the fact that direct-developing frogs usually lay eggs on the forest floor, and thus are independent of water bodies (*Nunes-de-Almeida, Haddad & Toledo, 2021*). These species were generally more abundant and diverse in the highland areas, where lentic ponds are scarce. This pattern has also been reported in other studies examining altitudinal gradients (*Naniwadekar & Vasudevan, 2007*; *Siqueira, Vrcibradic & Rocha, 2025*). The milder temperature and higher humidity should ensure additional protection against desiccation of their eggs deposited in the environment, contributing to the increased reproductive success of these species at higher altitudes. In contrast, leptodactylids and microhylids had one species registered for each family. Some anurans in the Atlantic Rainforest (*e.g.*, leptodactylids) build foam nests close to ponds to deposit and incubate egg clutches, which will hatch into tadpoles and then metamorphose into frogs (*Haddad & Prado, 2005*). At the same time, other species (*e.g.*, microhylids) present explosive reproduction, fossorial or semi-fossorial behavior, and low dispersal capacity in the environment, characteristics that make it difficult to sample adults in the field (*Peloso et al., 2014*; *De Sá et al., 2019*). Thus, perhaps this might be the reason for the low abundance of leptodactylids and microhylids that were restricted to lowland areas, as well as most of the species that use lentic ponds for breeding. The steepness of the terrain may hinder the formation of lentic ponds along the altitudinal gradient, decreasing the availability of suitable sites for the species that breed in these environments. Lastly, Hylidae was the most diverse family across the altitudinal range, with 21 species (58%) recorded. Neotropical anuran communities typically exhibit high hylid diversity (*Duellman, 1988*), a pattern reflected in both montane regions (*Carvalho-Rocha, Peres & Neckel-Oliveira, 2021*; *Matavelli et al., 2022*; *Santos-Pereira & Rocha, 2025*) and lowland areas of the Atlantic Forest (*e.g.*, *Mira-Mendes et al., 2018*). The dominance of hylids might be associated with mountain streams or aerial aquatic habitats (*e.g.*, tank-bromeliads) for developing eggs and tadpoles. Since reproductive modes are essential to understanding the distribution of anurans in altitudinal environments (*Siqueira et al., 2021*), these results might be directly associated with such reproductive strategies.

We found no statistically significant relationship between altitude and the different dimensions of alpha-diversity—taxonomic, phylogenetic, and functional. This suggests that, at least within the evaluated elevation range and transects, species richness and community structure remain relatively consistent along altitudinal gradient, supporting similar ecosystem functions and maintaining a comparable number of associated lineages and functional groups. This stability is likely maintained through functional redundancy,

where functionally equivalent species occupying similar niches at different altitudes compensate for changes in species composition. High-altitude environments often exhibit a reduction in taxonomic and functional diversity (*Graham et al., 2014*; *Villacampa et al., 2019*; *Zhao et al., 2022*; *Siqueira, Vrcibradic & Rocha, 2025*), which may lead to the loss of specific ecological functions. However, highlands are also characterized by microendemism, with species exhibiting unique traits—such as specialized reproductive modes or distinct physiological adaptations—that can partially compensate for reduced taxonomic diversity in terms of functional contributions to the ecosystem, while simultaneously enhancing phylogenetic diversity in these regions through the presence of distinct evolutionary lineages.

## Role of local environmental variables

Mountainous regions, despite covering only ~25% of the Earth's land surface, play a critical role in sustaining global biodiversity, harboring more than 85% of the world's species of amphibians, birds, and mammals, many of which are endemic to these environments (*Rahbek et al., 2019*). In the Atlantic Forest, different mountains are considered hotspots of anuran diversity (*e.g.*, *Forlani et al., 2010*; *Dias et al., 2014*; *Roberto & Loebmann, 2016*), but little is known about the main drivers of the anuran communities. Our results support that the number of tank-bromeliads plays an important role in anuran abundance, phylogenetic and functional diversity. The physical structure of some bromeliads enables rainwater to accumulate in the central tank and leaves axils, creating an important microhabitat for some anuran species from a wide variety of families (*Peixoto, 1995*; *Juncá & Borges, 2002*; *Lehtinen, 2004*; *Sabagh, Ferreira & Rocha, 2017*; *Zocca, Ghilardi-Lopes & Ferreira, 2024*). This complex architecture provides microhabitats for a diverse range of organisms, serving as sites for refuge, foraging, and even development (*Rocha et al., 2000*; *Lopez, Alves & Rios, 2009*). Some frog species, such as those in the genus *Phyllodytes*, spend their entire lifecycle within bromeliads (*Peixoto, 1995*). In the present study, six species of *Phyllodytes* were recorded, underscoring the importance of bromeliads as a critical resource for specialized anurans. Additionally, some species (*e.g.*, *Bokermannohyla lucianae*, *Pristimantis* sp. 2) were observed using bromeliads as vocalization sites in the study area. The role of the number of bromeliads in the anuran community was already highlighted in other studies (*e.g.*, *Bastazini et al., 2007*; *Silva, Carvalho & Bittencourt-Silva, 2011*; *Sabagh, Ferreira & Rocha, 2017*).

We found a positive association between leaf litter depth in the transects and the number of dominant lineages ($^2$PD). Transects with greater leaf litter depth had an equitable distribution of abundances among lineages. Previous studies have reported high anuran species richness associated with leaf litter, with species from different families utilizing litter verticality in distinct ways (*e.g.*, *Siqueira et al., 2009*; *Bruscagin et al., 2014*; *Rievers, Pires & Eterovick, 2014*). Thus, greater leaf litter depth may facilitate the coexistence of a higher number of species from different lineages that rely on this microhabitat. In contrast, in environments with shallower leaf litter, spatial limitations could increase competitive pressures, potentially leading to the local exclusion of certain lineages or reduction of their abundances (*Ovaskainen, Knegt & Mar-Delgado, 2016*),

particularly for those lineages that depend not only on the litter surface but also on its vertical structure. Additionally, moisture levels in the leaf litter may act as a limiting factor, influencing species distribution. Deeper layers retain higher humidity, providing a stable microhabitat for species dependent on moist environments. In contrast, shallow leaf litter tends to be drier, which may restrict the presence of many species. This could explain why only a subset of species is commonly found in shallow leaf litter environments. These species may exhibit distinct dehydration and rehydration rates compared to other terrestrial anurans, enabling them to thrive in drier, less humid conditions (*Dabés et al., 2012*). Thus, leaf litter moisture can act as an environmental filter, shaping anuran community distribution.

In addition to the availability of tank bromeliads, leaf litter and reproductive modes, temperature is often considered a key factor shaping anuran community structure along altitudinal gradients. Amphibians are highly dependent on climate due to their ectothermic metabolism, making temperature a key factor shaping geographic distribution and diversity patterns (*Hopkins, 2007*; *Duellman & Trueb, 1994*). Temperature decreases with elevation, and these two variables are highly correlated ($r^2 = 0.97$ in this study). A ~3 °C difference in air temperature was observed between lowland areas and the summit of Serra Bonita. The thermal variation establishes physiological stress gradients that act as environmental filters, limiting species occurrence to altitudinal ranges within their thermal tolerances. Consequently, species adapted to warmer lowland climates are gradually replaced by those specialized in colder, more humid conditions at higher elevations. This pattern reflects niche partitioning and the sensitivity of anurans to thermal fluctuations, underscoring the role of abiotic factors in structuring montane communities. Evidence from other studies in tropical altitudinal gradients suggests that temperature is a key driver of beta diversity in anuran communities (*Amador, Soto-Gamboa & Guayasamin, 2019*; *Carvalho-Rocha, Peres & Neckel-Oliveira, 2021*; *Matavelli et al., 2022*; *Bassetto et al., 2024*). However, in our study, we found no evidence that air temperature contributes more to community structuring than other local variables. Future studies incorporating temperature measurements across different microhabitats may provide a more refined understanding and help explain the different levels of amphibian diversity.

## Species composition and turnover along the altitudinal gradient

Despite the lack of significant associations between species richness, community structure, and environmental variables, we observed approximately 60% taxonomic dissimilarity between transects. This dissimilarity was primarily driven by altitudinal variation, indicating that while species richness and community structure remain relatively constant across elevations, species composition shifts along the altitudinal gradient. Similar patterns have been reported for other amphibian communities in tropical regions (*Amador, Soto-Gamboa & Guayasamin, 2019*; *Villacampa et al., 2019*). The results for phylogenetic diversity followed a similar pattern, although with a more pronounced effect above 600 m elevation. While lineage richness and the number of dominant lineages were not linked to elevation but rather to local environmental factors (specifically, number of tank-bromeliads and leaf litter depth), variations in lineage composition among transects

were significantly driven by elevation and marginally influenced by geographic distance. These findings are consistent with (*Wang et al., 2022*), who reported a positive relationship between phylogenetic dissimilarity and elevational divergence. Likewise, *Azevedo et al. (2021)* found that amphibian phylogenetic dissimilarity increases monotonically over distances of 0–500 km, a considerably greater distance than that analyzed in this study, yet consistent with our findings. We found no significant relationship between functional dissimilarity and environmental variation across transects, suggesting that the same ecological functions are maintained along the environmental gradient, regardless of local differences. These findings contrast with those of *Wang et al. (2022)*, who reported an increase in functional dissimilarity with elevation. However, their study covered a gradient from 0 to 2,200 m, whereas the variation in functional dissimilarity in our study may not have been broad enough to detect a significant association with any variable.

We observed that the anuran composition was influenced by the altitudinal gradient in which some frogs were found at specific altitudes. For instance, *Rhinella hoogmoedi*, *Dendropsophus novaisi*, *Trachycephalus nigromaculatus*, and *Physalaemus erikae* were restricted to the lowest altitudinal bands; *Chiasmocleis crucis* and *Bahius bilineatus* occurred just at mid-altitudes; and *Brachycephalus pulex*, *Ischnocnema verrucosa*, *Pristimantis* sp. 1, *Gastrotheca pulchra*, *Aplastodiscus ibirapitanga*, *A. weygoldti*, *Bokermannohyla lucianae*, and *Ololygon strigilatus* occurred only at higher altitudes. Other species were found throughout the altitudinal range (see Fig. 4). Species composition changes in anuran communities in response to altitudinal gradients have already been reported for different mountains worldwide (*e.g.*, *Hu et al., 2011*; *Zancolli, Steffan-Dewenter & Rödel, 2014*; *Matavelli et al., 2022*). Environmental conditions in montane ecosystems vary across the range (*Lomolino, 2001*; *Tito, Vasconcelos & Feeley, 2020*), and therefore, different environmental filters might influence the species composition.

## Rapoport altitudinal rule

Our results did not corroborate Rapoport's altitudinal rule, in which range sizes increased with altitude (*Stevens, 1992*). Instead, we observed an inverse pattern where anurans, with their midpoints at lower elevations, tend to cover broader elevational range sizes. Although supported in some studies (*Chen et al., 2020*; *Matavelli et al., 2022*), the anurans' distribution in altitudinal gradients seems to be inconsistent with the original predictions of Rapoport's rule (*e.g.*, *Goyannes-Araújo et al., 2015*; *Khatiwada et al., 2019*; *Siqueira et al., 2021*; *Dahl et al., 2024*). Our results suggest that species with optimal altitudes in the highlands exhibit a lower range-size distribution. In particular, this may reflect a habitat specialization or microendemism at higher altitudes (*Siqueira et al., 2021*). In addition, anurans' occurrence and abundance might be associated with environmental characteristics (*Almeida-Gomes, Rocha & Vieira, 2016*; *Araújo, Guzzi & Ávila, 2018*; *Pereira-Ribeiro et al., 2020*, this study). However, considering local and global scales, further studies are still needed to understand the principal filters driving anuran distribution in montane ecosystems.

## Conclusions and implications for conservation

Future projections indicate that climate change may significantly impact the functional and phylogenetic diversity of amphibians in lowland regions of the Atlantic Forest, driving species migration toward higher-altitude climate refuges (*Lourenço-de-Moraes et al., 2019*). Although our study found relative stability in functional diversity along the altitudinal gradient—suggesting that functional redundancy may buffer montane communities from immediate climate-driven declines—this resilience may be temporary. Over time, the influx of new species into higher elevations could alter competitive dynamics and disrupt ecosystem stability. These findings underscore the urgent need to conserve high-altitude areas in the region to preserve evolutionary potential and maintain ecosystem resilience in the face of ongoing climatic shifts. Additionally, implementing long-term monitoring programs in these high-altitude areas is essential to track the impacts of species migrations, assess changes in community composition, and detect emerging threats. Such efforts would provide critical data to inform adaptive conservation strategies and ensure the protection of these vital refuges in a rapidly changing climate.

The drivers of anuran distribution in Neotropical forests are influenced by local factors. In our study area, we observed a high abundance of bromeliads at higher elevations, and our analyses identified them as important drivers of diversity. Most areas of the Atlantic Forest reveal a lower bromeliad diversity, which could be why they have yet to be pointed out as the main diversity drivers in other areas (*Paz et al., 2020*). Additionally, our study also identified leaf litter depth as a positive driver of lineage evenness in the study area. Our findings support the notion that local factors, such as the presence of bromeliads and leaf litter, contribute to maintaining diverse amphibian lineages and functions within tropical forests. This insight may help identify key factors for amphibian conservation in disturbed landscapes, agroforestry systems, and areas undergoing ecological restoration. Understanding anurans' distribution and diversity patterns along altitudinal gradients and local factors that promote the maintenance of these patterns and diversity is essential to establish effective and targeted actions for conserving this taxonomic group. In addition, inventories with efforts directed only at forest environments can evidence a high species diversity and reveal the presence of little-known species and restricted endemics (*e.g.*, *Brachycephalus pulex*, *Pristimantis* spp., *Ischnocnema* spp.). In the mountainous complex of Serra Bonita, any conservation initiative should prioritize habitat protection in both lowland and highland areas, as each altitudinal stratum has a unique species composition.

## ACKNOWLEDGEMENTS

We extend our gratitude to Vitor Becker and Clemira Souza for their assistance during fieldwork. We also thank all the staff at the Uiraçu Institute, especially the park rangers, for their support. We are grateful to Rachel Pinto for her help in preparing the map.

### Funding

Iuri Ribeiro Dias, Marcelo Napoli, and Mirco Solé were supported by productivity grants from the Conselho Nacional de Desenvolvimento Científico e Tecnológico (CNPq) (Processes: 315362/2021-9, 314496/2021-1, and 309365/2019-8, respectively). Caio Vinicius de Mira-Mendes was funded by a Senior Productivity Grant (05/2023–PPG/UEMA) from the Universidade Estadual do Maranhão (UEMA). Kassio Castro Araújo received a research fellowship from CNPq and the Fundação de Amparo à Pesquisa do Estado do Piauí (FAPEPI) (Process: 150013/2023-0). This work was part of the project "Characterization of terrestrial vertebrates at the RPPN Complex of Serra Bonita as an aid for its effective management" supported by the Boticário Group Foundation for Nature Protection (Project No. 0818_20091). The funders had no role in study design, data collection and analysis, decision to publish, or preparation of the manuscript.

### Grant Disclosures

The following grant information was disclosed by the authors:
Conselho Nacional de Desenvolvimento Científico e Tecnológico (CNPq): 315362/2021-9, 314496/2021-1, 309365/2019-8.
Senior Productivity Grant: 05/2023–PPG/UEMA.
Universidade Estadual do Maranhão (UEMA).
CNPq.
Fundação de Amparo à Pesquisa do Estado do Piauí (FAPEPI): 150013/2023-0.
Boticário Group Foundation for Nature Protection: 0818_20091.

### Competing Interests

The authors declare that they have no competing interests.

### Author Contributions

- Iuri Ribeiro Dias conceived and designed the experiments, performed the experiments, analyzed the data, prepared figures and/or tables, authored or reviewed drafts of the article, and approved the final draft.
- Kássio de Castro Araújo analyzed the data, prepared figures and/or tables, authored or reviewed drafts of the article, and approved the final draft.
- Jorge Mario Herrera-Lopera analyzed the data, prepared figures and/or tables, authored or reviewed drafts of the article, and approved the final draft.
- Caio Vinícius de Mira-Mendes conceived and designed the experiments, authored or reviewed drafts of the article, and approved the final draft.
- Tadeu Teixeira Medeiros performed the experiments, authored or reviewed drafts of the article, and approved the final draft.
- Marcos Ferreira Vila Nova performed the experiments, authored or reviewed drafts of the article, and approved the final draft.

- Marcelo Felgueiras Napoli analyzed the data, authored or reviewed drafts of the article, and approved the final draft.
- Mirco Solé conceived and designed the experiments, authored or reviewed drafts of the article, and approved the final draft.

### Animal Ethics

The following information was supplied relating to ethical approvals (*i.e.*, approving body and any reference numbers):

This research was approved by the Ethical Committee on Animal Use at the Universidade Estadual de Santa Cruz (CEUA-UESC 006/09).

### Field Study Permissions

The following information was supplied relating to field study approvals (*i.e.*, approving body and any reference numbers):

Field experiments were approved by the Instituto Chico Mendes de Conservação da Biodiversidade—ICMBio (#13708).

### Data Availability

The raw data and R code are available in the Supplemental Files.

### Supplemental Information

Supplemental information for this article can be found online at http://dx.doi.org/10.7717/peerj.19561#supplemental-information.

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
