# Peer review of "From lowlands to highlands: how elevation and habitat complexity drive anuran multidimensional diversity?"

_PeerJ, doi:10.7717/peerj.19561_

## Round 0.1 · original submission · Major Revisions

Please respond to all the comments by the reviewers when you submit your revised manuscript. I would urge you to pay special attention to the detailed comments provided by Reviewers 2 and 3. I look forward to receiving your revised manuscript.

Reviewer 1 ·

Basic reporting

Basic reporting is very good. The Ms is clearly written. However, please be aware that myself I am not a native English speaker, therefore I might not be have the best expertise in judging the English language!
I suggest to review the Ms for redundancies. In some parts (in particularily in the discussion) there are some unnecessary redaundancies, basically the same point is made several times just in different ways, such as in the discussion about Rapoprt's rule (lines 333-340). Please check the ms thoroughly for redundancies.
Background and context of the study is clearly presented.
The cited literature is adequate. Here, I would recommend to include some more studies that are NOT fom the same study area or the from the Brazilian Atlantic forest. The topic of teh research is certainly of high interest to a more general public, and there are studies on altitude & biodiversity from many parts of the world. And also, given some information on animal taxa that are not amphibians would be great; and then to discuss the patterns found in amphibians a bit more in respect tp patterns found in other animal taxa; not extensively, but a bit more...
Figures are good, raw data is shared.
The study has relevant results to the hypotheses.

Experimental design

The study is clearly primary research, and fits the ais and the scope of PeerJ.
The research question is well defined, relevant and interesting. As a comment: I don't see that species richness and distribution in the mountains remains an "unsolved enigma.", this expression (the "enigma" part) seems a bit exaggerated, it is just a bit complicated.
In general, the study is methodologically very well performed, both the field work (creating a certainly sufficient data set) and the analytical part. The meothds, again both field work and statistics, are very well explained (if the Appendix is also included).

Validity of the findings

All relevant data have been shared.
I consider the statistics as appropriate, and well-performed. The methods section shows (and in particular also the extended methods in the Appendix) that the athors have very thoroughly applied first exploratory statistics, and have very thoroughly tested first if all requirements for certain statistical tests were met. As a side comment: I haven't seen using PC-Ord for quite a long time... nice, it's a great program!
One point that I ask the authors to check is about the one-dimensional NMDS. I have performed NMDS with mostly two, sometimes three dimensions, where the stress value should be below 20 (0,20) to be ok that the NMDS solution is really including the multi-dimensional variability in the data. There are some stress values that are much higher (>30) in this study. The authors actually test for differences in stress between their data and a null model, I see this, and the "threshold" of stress for the 1-dimensional solution might be different, but the authors might check this.
Conclusions well stated. As I mentioned above, there are some redundancies that could be eliminated, and I would reocmmend to put the study in a bit wider context (away from the Atlantic Forest and from frogs, i.e. including otehr systems both geographically and taxonomically). However, as a summary, I consider this study as interesting and well-done, and I would like to see it published.

·

Basic reporting

The study examines amphibian diversity along an altitudinal gradient in the Atlantic Forest of northeastern Brazil, identifying 36 anuran species from 10 families and emphasizing the critical role of elevation in shaping anuran community composition, richness, and abundance. Environmental variables, particularly the density of tank-bromeliads—a vital habitat for many amphibians—were found to significantly influence anuran distributions. This research provides valuable insights into the distribution, abundance, and composition of amphibian species in the northeastern Brazilian forest and underscores the importance of understanding ecological drivers for effective conservation strategies. While the manuscript is generally well-written, the methods section requires further clarification.

Experimental design

Here are some specific comments and suggestions:

1. The phrase "Scaling the Heights" in the title might be misleading since the study does not address scaling issues or employ a spatially hierarchical sampling design. It would be advisable to remove this phrase.
2. In Figure 1, the four altitudinal bands are clearly depicted. However, for the six line transects, it would be more accurate to show the actual survey routes rather than hypothetical straight lines. Ideally, the survey routes should be overlaid on fine satellite imagery, and the exact locations of species occurrences should be plotted along these routes.
3. At line 157, the statement "The species abundance in each transect was determined by the highest recorded abundance..." suggests a method that does not fully utilize the collected data. Instead of relying solely on peak abundances, consider using the original abundance data for each survey, incorporating months as a random effect in a mixed-effects model or treating months as a categorical variable in a linear model (ANCOVA) to capture temporal variability.
4. At line 170, where it states "excluded the variable altitude from the GLM models (VIF > 10)...", given that the study focuses on altitudinal gradients, excluding altitude due to multicollinearity seems counterintuitive. Instead of removing the variable with the highest VIF, consider eliminating another less critical variable to reduce VIF values until they fall below the threshold of 5 or 10.
5. Lines 171-172. The model is presented here. I strongly suggest adding quadratic terms and two-way interaction terms in the model, followed by a model selection process to remove terms with low effects. Including a quadratic term can indicate that the most suitable conditions often occur at moderate levels. Interaction effects are also frequently important.
6. At line 173, stating "family = Gaussian" indicates that the model used is a general linear model rather than a generalized linear model. This is appropriate for NMS coordinates and abundance but may not be suitable for richness, which might follow a Poisson or negative binomial distribution. If richness exhibits such a distribution, Poisson regression or negative binomial regression should be used instead.
7. At line 194, the statement "We give a range of 100 m to species recorded at a single sampling point" is unclear. It would be more precise to use the exact altitude of each occurrence. A density plot showing the altitudinal range of all occurrences would be the best way to represent a species' altitude niche, illustrating the minimum, maximum, and mode of the variable.
8. Fig. 5 is a very good figure!

Validity of the findings

No coments.

Additional comments

No coments.

·

Basic reporting

Dear Editor,

In the manuscript: "Scaling the Heights: Understanding Frog Diversity Along the Altitudinal Gradient of Southern Bahia's Atlantic Forest", the authors described the anuran composition, species richness, and distribution patterns along an altitudinal gradient in the Atlantic Forest of Brazil and how environmental variables explain these patterns. The manuscript is well written, but I think that based on the dataset that the authors have, other analyses can be performed and improve the results, discussion, and manuscript in general. I recommend trying to explore taxonomic and phylogenetic diversity and compare it with species richness among the different sampling sites. I leave other comments in the Word version of the manuscript.

Experimental design

Other analyses can be performed based on the dataset, for example taxonomic and phylogenetic diversity.

Validity of the findings

Based on the current analyses, the results are accurate.

---

## Round 0.2 · Minor Revisions

I am happy that both reviewers like the revised manuscript. Please address the comments by Reviewer 1 and resubmit your manuscript.

Reviewer 1 ·

Basic reporting

As in the previous version, good.

Experimental design

As in the previous version, good.

Validity of the findings

As in the previous version, good.

Additional comments

This is the revised version of the manuscript, and I am not going into detail for all aspects, since I have done this before. The authors have done a very good job revising their manuscript. In particular, including phylogenetic and functional diversity of the frogs communities as response variables very much improved the relevance of the study; well done! I do have just one more specific comment, please see below. In summary: I see this as an interesting, well-performed and well-written study, that I would be happy to see published.

Lines 503 to 529: My comment refers mainly to the wording (NOT the usage!) of the model selection via AIC, here in this part and elsewhere (e.g. discussion). This wording seems to be misleading in some parts. Model selection via AIC (or here, AICc) already includes an approach partly based on parsimony. AIC already combines explanatory power (e.g. RSS) AND the number of variables in the model, i.e. the less variables in a model, the lower (i.e., “better”) the AIC.
For example: “…Although alternative models included explanatory variables with significant associations, their explanatory power was not substantially better than that of the null model (ΔAICc < 2, Figs. 2C-E)…”. In my understanding, you cannot show differences in explanatory power between two models with the argument that the AIC is similar, as the AIC is equally influenced by the number for variables in the respective model, and the number of variables is obviously different when comparing a null model with some other model. Also, the statement “…the null model was the most parsimonious…” is true but a null model is always the most parsimonious model, in my understanding.
There might be a chance that I am wrong here, but I suggest that the authors carefully check through their statements dealing with AIC, parsimony, and explanatory power.

·

Basic reporting

Dear Editor and authors,

Regarding to the manuscript "From lowlands to highlands: How elevation and habitat complexity drive anuran multidimensional diversity?" I am happy and satisfied with all the work that authors made to comply with the comments and suggestions made by the reviewers. Specially I am very impressed with the work made to include phylogenetic and taxonomic diversity analysis in their research. I thank the authors have included the raw data and scripts in a public repository.

Experimental design

No comment

Validity of the findings

No comment

Additional comments

Dear Editor and authors,

Regarding to the manuscript "From lowlands to highlands: How elevation and habitat complexity drive anuran multidimensional diversity?" I am happy and satisfied with all the work that authors made to comply with the comments and suggestions made by the reviewers. Specially I am very impressed with the work made to include phylogenetic and taxonomic diversity analysis in their research. I thank the authors have included the raw data and scripts in a public repository.

---

## Round 0.3 · accepted · Accept

Thank you for addressing the concern of the reviewer and making changes in your model selection text. This makes the process more clear.